# Nerve regeneration using a Bio 3D conduit derived from umbilical cord–Derived mesenchymal stem cells in a rat sciatic nerve defect model

Terunobu Iwai[1], Ryosuke Ikeguchi[1,2]*, Tomoki Aoyama[3], Takashi Noguchi[1], Koichi Yoshimoto[1], Daichi Sakamoto[1], Kazuaki Fujita[1], Yudai Miyazaki[4], Shizuka Akieda[4], Tokiko Nagamura-Inoue[5], Fumitaka Nagamura[6], Koichi Nakayama[7], Shuichi Matsuda[1]

1 Department of Orthopaedic Surgery, Kyoto University, Kyoto, Japan, 2 Department of Rehabilitation Medicine, Kyoto University, Kyoto, Japan, 3 Human Health Sciences, Graduate School of Medicine, Kyoto University, Kyoto, Japan, 4 Cyfuse Biomedical K.K., Tokyo, Japan, 5 Department of Cell Processing and Transfusion, The Institute of Medical Science, IMSUT CORD, Research Hospital, The University of Tokyo, Tokyo, Japan, 6 Division of Advanced Medicine Promotion, The Advanced Clinical Center, The Institute of Medical Science, The University of Tokyo, Tokyo, Japan, 7 Department of Regenerative Medicine and Biomedical Engineering, Faculty of Medicine, Saga University, Saga, Japan

* ikeguchir@me.com

**Data Availability Statement:** All relevant data are within the manuscript.

## Abstract

Human umbilical cord–derived mesenchymal stromal cells (UC-MSCs), which can be prepared in advance and are presumed to be advantageous for nerve regeneration, have potential as a cell source for Bio 3D conduits. The purpose of this study was to evaluate the nerve regeneration ability of Bio 3D conduits made from UC-MSCs using a rat sciatic nerve defect model. **Methods:** A Bio 3D conduit was fabricated using a Bio 3D printer by placing UC-MSC spheroids into thin needles according to predesigned 3D data. The conduit was transplanted to bridge the 5-mm gaps of Lewis rat sciatic nerve, and nerve regeneration was evaluated at 8 weeks (Bio 3D group). Transplantation of autologous nerve segments (autograft) and silicone tubes represented the positive and negative control groups, respectively. In a second experiment, immunological reactions were evaluated in Bio 3D, autograft, and allograft groups by histochemical staining of transplanted segments in Brown Norway rats. **Results:** The mean angle of attack value in the kinematic analysis was significantly better in the Bio 3D group ($-20.1 \pm 0.5°$) than in the silicone group ($-33.7 \pm 1.5°$) 8 weeks after surgery. The average diameters of myelinated axons were significantly larger in the Bio 3D group ($3.61 \pm 0.15$ μm) than in the silicone group ($3.07 \pm 0.12$ μm), and the number of myelinated axons was significantly higher in the Bio 3D group ($11,201 \pm 980$) than in the silicone group ($8117 \pm 646$). Histological findings (hematoxylin and eosin [HE] staining and anti-CD3 fluorescent immunostaining) showed that rejection was suppressed in the Bio 3D group compared to the allograft group. Based on macroscopic findings and histological findings (anti-human mitochondrial fluorescent immunostaining), UC-MSCs in the Bio 3D conduit disappeared gradually from week 1 to week 8. **Conclusions:** The Bio 3D conduit prepared from UC-MSCs was superior to the silicone tube and achieved comparable nerve

**Funding:** This study was supported by Japan Agency for Medical Research and Development, Grant Number 22bk0104155h0001.

**Competing interests:** There are no patents or marketed products to declare. KN is the co-founder and shareholder of Cyfuse Biomedical K.K., Tokyo, Japan (Cyfuse). YM and SA, who are employees of Cyfuse, contributed to the manufacturing of 3D conduits and Cyfuse provided the bioprinter to manufacture the conduit. The company has the industrial rights related to the bioprinting method used to construct the 3D conduit in this work. Cyfuse provided support in the form of salaries for authors YM, SA and KN and provided research grants to TA, KN and SM. These competing interests do not alter the authors' adherence to PLOS ONE policies on sharing data and materials.

regeneration to the autologous (autograft) group. Rejection was suppressed in the Bio 3D group compared to the allograft group. Although this study used a xenograft model, we speculate that rejection was low due to the characteristics of UC-MSCs. UC-MSCs are a useful cell source for Bio 3D conduits.

## Introduction

Peripheral nerve injury can lead to severe and long-term impairment of activities of daily living and quality of life. The mainstream treatment for peripheral nerve defect is autologous nerve grafting, which has good potential for nerve recovery [1–4]. Artificial nerve conduits have been developed to avoid sacrificing the donor's healthy nerve. However, nerve regeneration via artificial nerve conduits is suboptimal compared with that achieved through autologous nerve grafting. In our previous studies, we transplanted Bio3D conduits made from human fibroblasts into immunodeficient rats and observed good nerve regeneration [5–8]. However, for clinical application, the process of collecting and culturing fibroblasts before conduit transplantation takes time. Human umbilical cord–derived mesenchymal stem cells (UC-MSCs) have the advantage of storage stability, as they can be frozen along with the bladder tissue, and are immune tolerant [9]. Compared to fibroblasts as a material for Bio3D conduits, UC-MSCs do not require tissue collection from the patient and the culture period can be shortened by about 3 weeks. In the current study, we transplanted human-derived UC-MSCs as material for a Bio3D conduit into a rat sciatic nerve model without immunosuppression and evaluated the effectiveness of nerve regeneration.

This study consisted of two parts, evaluating (1) nerve regeneration, and (2) acute nerve rejection. In the first experiment, Bio 3D conduits made from UC-MSCs were transplanted using a rat sciatic nerve 5-mm defect model (Bio 3D group); nerve regeneration was evaluated and compared with that of the autograft control group, in which a 5-mm autologous nerve segment was transplanted, and the silicone control group, in which a silicone tube was transplanted to a 5-mm nerve defect. In the second experiment, to evaluate acute graft rejection, we compared the Bio 3D group with autograft and allograft groups (in which nerve segments harvested from Brown Norway rats were transplanted to Lewis rats) 1 week after surgery. We also confirmed that a number of transplanted UC-MSCs in the Bio 3D conduit remained 8 weeks after surgery.

## Materials and methods

### Bio 3D conduit

UC-MSCs were provided by IMSUT CORD (Cord Blood and Cord Bank) in The Institute of Medical Science, The University of Tokyo, Japan. IMSUT CORD activity was reviewed and approved by the IRB (No. 35–2). Conduits were fabricated from UC-MSCs using a Bio-3D printer (Regenova, Cyfuse Biomedical, Tokyo, Japan) using the methods described by Ito et al. [10].

Briefly, cells detached by trypsin treatment were centrifuged, and the number of cells in the suspension was counted. Then, the cells were resuspended at a concentration of $3 \times 10^5$ cells/mL and incubated in a low-cell-adhesion 96-well plate (Sumilon PrimeSurface1, Sumitomo Bakelite, Tokyo, Japan) for 24 h. Cells aggregated to form multicellular clusters called spheroids, each with a diameter of approximately $750 \pm 50$ μm.

A Bio 3D conduit was created using the Kenzan method [2]. The Bio3D printer aspirated spheroids from each well into a nozzle, placed them layer by layer into predesigned circularly arranged needles, and stacked them according to the predesigned pattern. Within a week of printing, the spheroids coalesced to create Bio3D conduits within the needle array. A Bio3D conduit was removed from the needle array and transferred to an 18-gauge intravenous catheter (Surflo; Nipro, Osaka, Japan). Conduit was grown in a perfusion bioreactor to the required size. The resulting bio3D conduit had an inner diameter of 2 mm, a wall thickness of 500 μm, and a length of 8 mm.

## Experimental group

All experiments were conducted in accordance with the guidelines of the Kyoto University Graduate School of Medicine Animal Care and Use Committee. Male Lewis rats (9–10 weeks old, 200–300 g; Japan SLC, Shizuoka, Japan) were used. Rats were housed in flat-bottomed cages in a pathogen-free room with ad libitum food and water. A total of 15 rats were randomly assigned into three groups of five rats each: groups transplanted with Bio 3D conduits, autologous nerves, and silicone tubes (Bio 3D, autograft, and silicone groups, respectively). Eight weeks after transplantation, nerve regeneration was evaluated (Fig 1A–1C).

In addition, an allogeneic nerve transplant group (allograft group) that used Brown Norway rats (9–10 weeks old, 200–300 g; Japan SLC) as nerve donors was prepared. The rejection reaction of nerves was evaluated 1 week after surgery, with three rats in each of the three groups: The Bio 3D, autograft, and allograft groups.

We also evaluated the persistence of transplanted UC-MSCs in the regenerated nerve in the Bio 3D group (n = 3) 8 weeks after transplantation compared with that in the Bio 3D group (n = 3) 1 week after transplantation and in the autograft group (n = 3) 1 week after transplantation.

## Surgical technique

Sciatic nerve surgery was performed under general anesthesia with isoflurane inhalation. With the rat in the prone position, a skin incision was made from the greater trochanter of the right lower limb to the posterior popliteal fossa. Then, the fascia of the gluteus muscle was incised, the muscle was divided, and the sciatic nerve was exposed.

In the Bio 3D group, a gap of 5 mm was created in the center of the right sciatic nerve, the proximal and distal stumps were drawn into the conduit at 1.5 mm each, and an 8-mm-long Bio 3D conduit was transplanted to bridge the nerve gap. Both stumps were secured with 10–0 nylon sutures under an operative microscope. In the silicone group, a 5-mm nerve gap was created, and an 8-mm silicone tube was used to bridge the nerve gap in the same manner as described above. For the autograft group, after exposing the right sciatic nerve, 5 mm of the nerve was excised, the proximal and distal stumps were reversed, and both stumps were sutured with 10–0 nylon suture. In the allograft group, 5 mm of a right sciatic nerve segment harvested from Brown Norway rats was sutured into a 5-mm nerve space created between the right sciatic nerves of Lewis rats. In all groups, the wound was closed with 5–0 nylon suture. For postoperative analgesia, 1 mg/kg meloxicam was administered intraperitoneally. Rats were euthanized at the indicated time points using carbon dioxide gas.

## Pinprick and toe-spread tests

Eight weeks after surgery, the pinprick and toe-spread tests were used to evaluate the functional recovery of sensory and motor nerves. These tests were developed by Siemionow et al. and the results were scored as grades 0 to 3 [11].

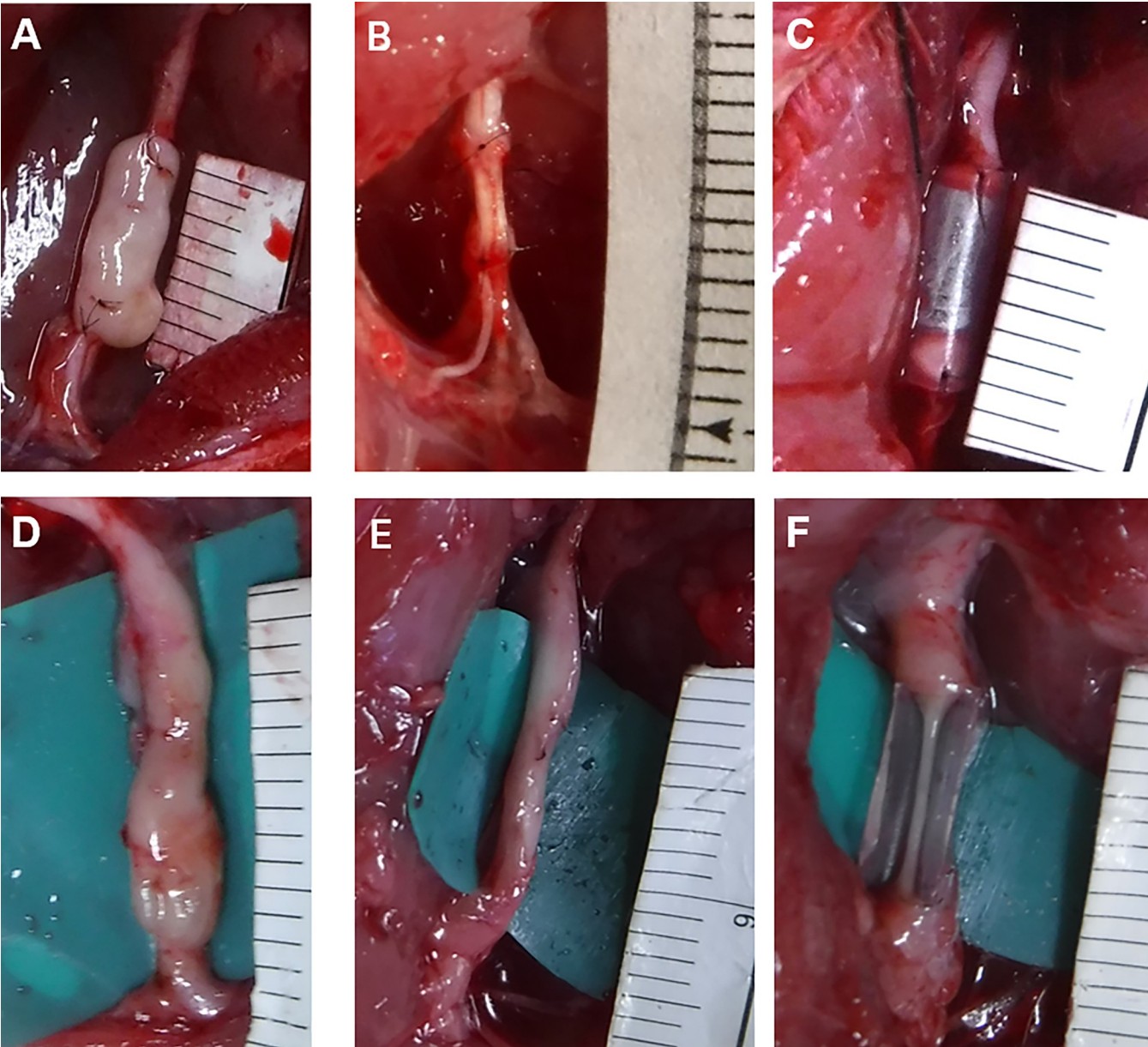

**Fig 1.** Three types of materials were transplanted into the 5-mm gaps of rat sciatic nerve: (A) Bio 3D conduit, (B) autograft conduit, and (C) silicone conduit. Regenerated nerves 8 weeks after surgery in the (D) Bio 3D group, (E) autograft group, and (F) silicone group. Nerves were regenerated in all three groups, but the regenerated nerves in the silicone group were thinner than those in the Bio 3D and autograft groups.

The pinprick test is used to evaluate the recovery of tactile and pain sensations. Rats were assessed by their response to having their paws clamped with forceps. Grade 0 indicates no response to stimulation, grade 1 indicates that the foot withdraws from stimulation to the heel, grade 2 indicates that the foot withdraws from stimulation to the top of the foot, and grade 3 indicates that the foot withdraws from stimulation to the toe. A higher grade indicates better recovery.

The toe-spread test is used to assess motor recovery. When rats are suspended by the tail, the toes normally abduct and hyperextend, but this movement is impaired when sciatic nerve paralysis is present. Grade 0 has no toe reaction, grade 1 has only slight toe movement, grade 2

involves toe abduction, and grade 3 involves toe abduction and extension. A higher grade indicates better motor recovery.

## Kinematic analysis

At 8 weeks after surgery, kinematic analysis was performed using a treadmill [12]. With the rat in the supine position, five ink-colored hemispherical plastic markers were placed on each joint of both lower limbs (anterior superior iliac spine, greater trochanter of the hip joint, stifle joint, ankle joint, and fifth metatarsophalangeal joint). Placement of the spherical plastic markers on the toes can interfere with walking, and the area where they can be attached is very limited, so acrylic resin ink was applied as a substitute for marking.

Treadmill walking was recorded by tracking each marker during walking using a 3D kinematic analysis system (Kinema Tracer System, Kissei Comtec, Nagano, Japan). We evaluated two parameters: (1) drag toe (DT), which is the percentage of walking that does not come off the ground and (2) angle of attack (AoA), which is the angle formed by the toe and metatarsal bone immediately before the toe touches the ground at the end of the swing phase (Fig 2A). A smaller DT value and a larger AoA value indicate better walking.

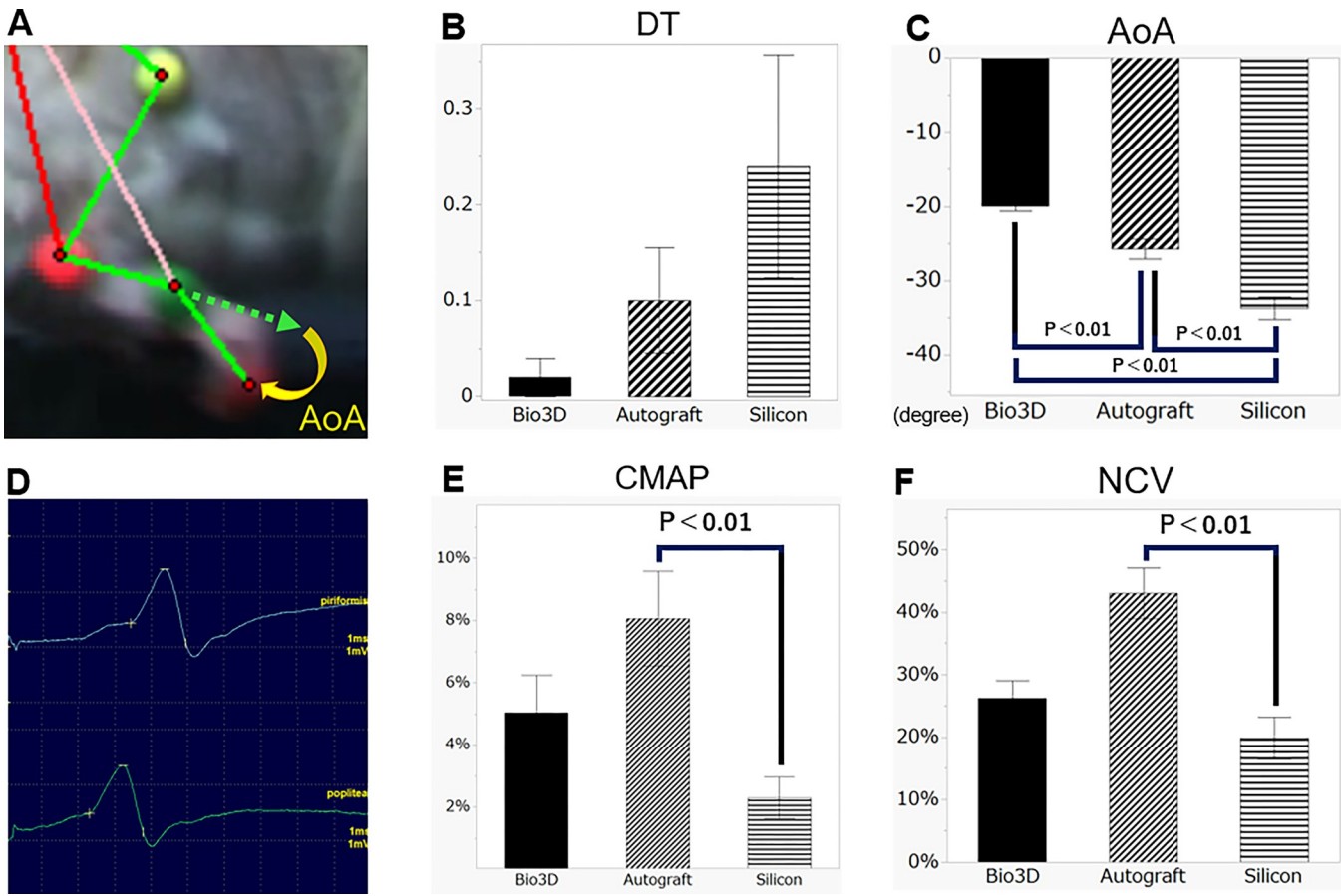

**Fig 2.** (A) Angle of attack (AoA) is formed by the toe and metatarsal bone immediately before the toe touches the ground at the end of the swing phase. (B) For DT, there was no significant difference among the three groups. (C) For AoA, the Bio 3D conduit performed significantly better than the autograft and silicone conduits. (D) The blue waveform is stimulated from the greater trochanter, and the green waveform is stimulated from the popliteal fossa (vertical frame: 1 ms, horizontal frame: 2 mV). (E) CMAP, and (F) NCV. For CMAP and NCV, the autograft was significantly better than the silicone conduit, and there was no significant difference between the Bio 3D conduit and the autograft. DT: Drag toe, AoA: Angle of attack, CMAP: Compound motor action potential, NCV: Nerve conduction velocity. Error bars represent the standard error.

## Electrophysiological studies

Nerve conduction studies were performed to measure the compound muscle action potential (CMAP) and nerve conduction velocity (NCV) of the adductor foot muscle, which is innervated by the sciatic nerve. Under general anesthesia with isoflurane, a point immediately posterior to the greater trochanter (S1) and popliteal fossa (S2) was detected using an evoked potential testing system (Neuropack S1, MEB-9404, Nihon Kohden, Tokyo, Japan) (Fig 2D). The treatment was performed by inserting a stimulating electrode into the sciatic nerve, and evoked potentials were obtained by inserting a pair of electrodes into the leg adductor muscles. CMAP was obtained by measuring the height (peak-to-peak) of waveforms evoked by stimulation of point S1. NCV was calculated by dividing the exact distance between points S1 and S2 by the latency difference between the two evoked potential waveforms evoked by stimulating points S1 and S2. The same procedure was performed on the healthy side, and the ratio of the operated side to the healthy side was calculated.

## Macroscopic observation

A skin incision was made at the same site as the previous surgery. The right sciatic nerve was exposed, and changes in the surgical site were visually observed to confirm nerve regeneration and tumor development.

## Wet weight of the tibialis anterior muscle

The tibialis anterior muscle on the operated side was dissected and collected, and the wet weight of the muscle was measured using a digital scale. Then, the ratio of the wet muscle weight of the operated side to that of the heathy side was determined.

## Morphological evaluation

The mid portion of the regenerated sciatic nerve on the operated side was fixed with 1% glutaraldehyde and 1.44% paraformaldehyde in all three groups. The samples were then fixed with 1% osmic acid and embedded in epoxy resin to allow 1-μm-thick sections to be cut laterally. The sections were stained with 0.5% (w/v) toluidine blue solution and the slides observed at 400× magnification using a light microscope.

## Number and dimensions of myelinated axons

Image J software (National Institutes of Health, Bethesda, MD, USA) was used to count the total number of myelinated axons in all nerve areas in each sample.

The properties of regenerated myelinated axons were analyzed from the same samples used to assess the total number of myelinated axons in each sample of the three groups. Ultrathin cross sections (1 μm) of the samples were stained with lead citrate and uranyl acetate, and then examined using a transmission electron microscope (TEM; model H-7000, Hitachi High Technologies, Tokyo, Japan) at a magnification of 2000×. The shortest myelinated axon diameter (a) and axon diameter (b) were measured using Image J software. From these two measurements, the thickness of the myelin sheath ([a − b]/2) in each field was calculated. The mean values of the three parameters were calculated in each field; from these results, we obtained the mean values in all view fields (6 view fields) in each sample of the three groups.

## Mononuclear cell infiltration

In the Bio 3D, autograft, and allograft groups, the right sciatic nerve was harvested 1 week after surgery and fixed with 4% paraformaldehyde. After embedding in paraffin, the samples were

sliced at a thickness of 5 μm and stained with HE. The slides were observed at a magnification of 400× using a confocal microscope (BZ-X700; Keyence, Osaka, Japan). We photographed six random view fields for each sample. and the number of mononuclear cells infiltrating nerves within the view field was counted.

## Immunohistochemical staining

Immunohistochemical staining was performed on samples from each group. Sections of the mid portion of the regenerated sciatic nerve were fixed with 4% PFA and then immersed in 20% sucrose solution, and the cryoprotected samples were embedded with OCT compound. The frozen samples were sliced horizontally and vertically to a thickness of 14 μm and embedded in glass slides. These prepared samples were washed with phosphate-buffered saline (PBS) and subjected to antigen retrieval with proteinase K (Sigma Aldrich, St. Louis, MO, USA) for 15 min at room temperature. We added primary antibodies: anti-CD3 antibody (1:150, Abcam, Tokyo, Japan) as a T-cell marker, FLEX Polyclonal Rabbit Anti-S100 (1:2, Dako, Carpinteria, CA, USA) as a Schwann cell marker, NF-H Antibody (RNF402) Mouse-Mono (1:50, Novus Biologicals, Centennial, CO, USA) as a neurofilament marker, and Anti-Mitochondria antibody (1:100, Abcam) as a human-specific marker. All antibodies were incubated for 24 h at 4˚C. Slides were washed with PBS and tested for anti-rabbit IgG (H+L), highly cross-adsorbed, CF™ 555 antibody produced in donkey, anti-mouse IgG (H+L), highly cross-adsorbed. CF™ 488A antibody produced in donkey (Sigma-Aldrich) was added as a secondary antibody and incubated at room temperature for 1 h. After washing with PBS, coverslips were mounted on slides using non-hardening Fluoro-Keeper anti-fade reagent with DAPI (Nacalai Tesque, Kyoto, Japan). In addition, a negative control slide was prepared to determine the exposure time to exclude nonspecific findings. These slides were observed using a confocal microscope (BZ-X700; Keyence). We photographed 10 random view fields for each sample in the immunohistochemical staining of anti-CD3 antibody and anti-mitochondria antibody staining. The number of bright spots within the view field was counted.

## Plasma cytokines

Concentrations of plasma cytokines interleukin (IL)-2, IL-10, tumor necrosis factor (TNF)-α, and interferon (IFN)-γ were measured to assess acute rejection and systemic inflammatory response in transplant-treated rats. Plasma samples from each rat were collected 1 week after transplantation [13]. Approximately 250 μL of blood was collected from the tail vein, and the separated plasma samples were stored at −80˚C. Plasma concentrations of IL-2, IL-10, TNFα, and IFN-γ were measured using enzyme-linked immunosorbent assay (ELISA) kits (R&D Systems, Minneapolis, MN, USA).

## Statistical analysis

Data are presented as mean values and standard error values. The Shapiro–Wilk test was performed to confirm whether each value was normally distributed. Data from kinematic analysis and electrophysiological study, the values of the wet muscle weight measurement, the number of myelinated axons and their measured properties (diameter of the axon and myelinated axon and myelin thickness) were compared using one-way analysis of variance, and a post hoc test was applied using the Tukey–Kramer test (JMP Pro software version 15.10.0; SAS Institute Inc., Cary, NC, USA). Data from mononuclear counts with HE staining and the number of bright spots in immunohistochemical stain and plasma cytokine was applied using the Steel–Dwass test. These values were considered statistically significant at p < 0.05 for all data.

## Results

### Pinprick test/toe-spread test

In the pinprick test, in the Bio 3D group, three rats showed grade 3 and two rats grade 2. In the autograft group, one rat was grade 3 and four rats were grade 2. In the silicone group, four rats were grade 2 and one rat was grade 1.

In the toe-spread test, in the Bio 3D group, two rats were grade 3 and three rats were grade 2. In the autograft group, three rats were grade 3 and two rats were grade 2. In the silicone group, two rats were grade 3 and three rats were grade 2.

There were no significant differences between groups in the pinprick test or the toe-spread test.

### Macroscopic observation

Eight weeks after transplantation, changes in the sciatic nerve transplantation site in each group were observed (Fig 1D–1F). Regeneration of the sciatic nerve was observed in all rats in the three groups. No tumor formation, including neuroma, was observed in any of the rats. The diameter of the regenerated nerve was smaller in the silicone group than in the Bio 3D and autograft groups.

### Kinematic analysis

The mean DT was $0.02 \pm 0.02$ in the Bio 3D group, $0.1 \pm 0.05$ in the autograft group, and $0.24 \pm 0.12$ in the silicone group. There was no significant difference between groups (Fig 2B).

The mean AoA values were $-20.1 \pm 0.5°$ in the Bio 3D group, $-25.7 \pm 1.4°$ in the autograft group, and $-33.7 \pm 1.5°$ in the silicone group. The AoA value of the Bio 3D group was significantly lower (i.e., better) than that of the autograft and silicone groups ($p = 0.0033$ and $p < 0.001$, respectively; Fig 2C).

### Electrophysiological studies

The mean ratio of CMAP of the adductor pedis was $5.0 \pm 1.2\%$ in the Bio 3D group, $8.1 \pm 1.5\%$ in the autograft group, and $2.3 \pm 0.7\%$ in the silicone group. The autograft group showed significantly better recovery than the silicone group ($p = 0.0129$). There was no significant difference between the Bio 3D group and the autograft group (Fig 2E).

The mean ratio of NCV was $31.8 \pm 5.1\%$ in the Bio 3D group, $43.1 \pm 4.0\%$ in the autograft group, and $19.8 \pm 3.4\%$ in the silicone group. The autograft group showed a higher (i.e., better) mean ratio than the silicone group ($p = 0.0055$). There was no significant difference between the Bio 3D group and the autograft group (Fig 2F).

### Wet weight of the tibialis anterior muscle

The mean ratio of the wet weight of the tibialis anterior muscle was $48.2 \pm 1.8\%$ in the Bio 3D group, $55.34 \pm 6.0\%$ in the autograft group, and $42.8 \pm 2.9\%$ in the silicone group (Fig 3A–3C). Significant differences were observed between the autograft group and the silicone group ($p = 0.0068$) (Fig 3D).

### Morphological evaluation

Ultrathin transverse sections revealed that myelinated axons with larger diameter and thicker myelin sheath were regenerated in the Bio 3D group and the autograft group compared with the silicone group (Fig 4A–4F). Quantitative results are as follows.

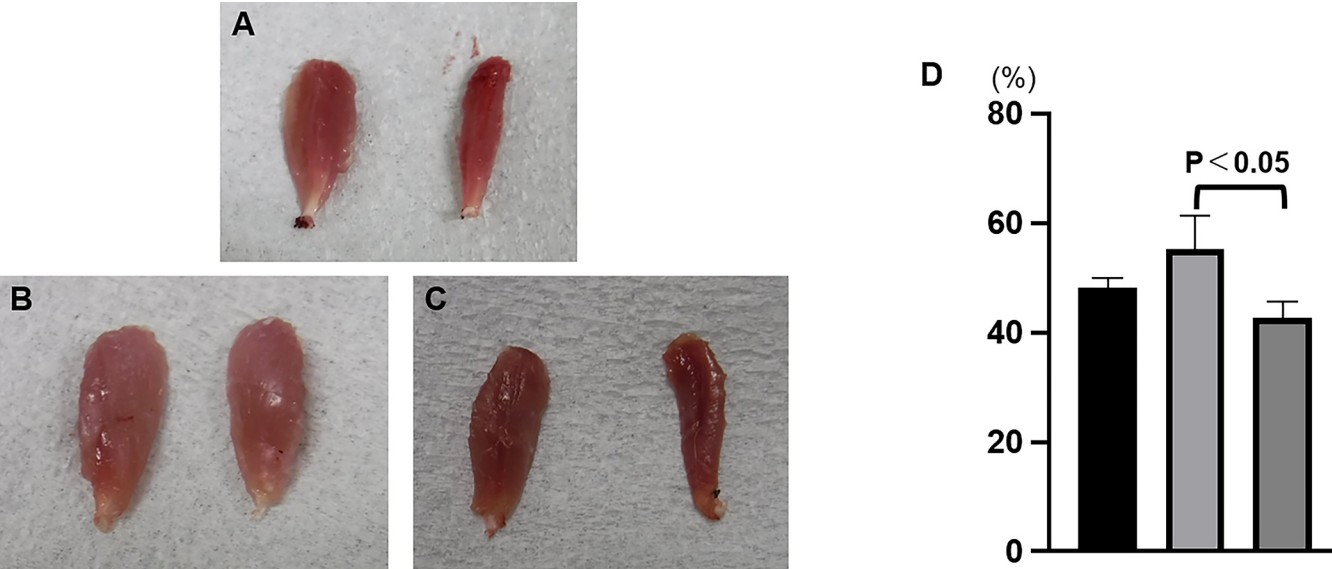

**Fig 3.** The tibialis anterior muscle on the right side is the healthy side, and the muscle on the left is the transplanted side: (A) Bio 3D group, (B) autograft group, (C) silicone group. (D) Based on the wet weight of the tibialis anterior muscle of the transplanted side, the autograft group showed significantly better recovery than the silicone group. Error bars represent the standard error.

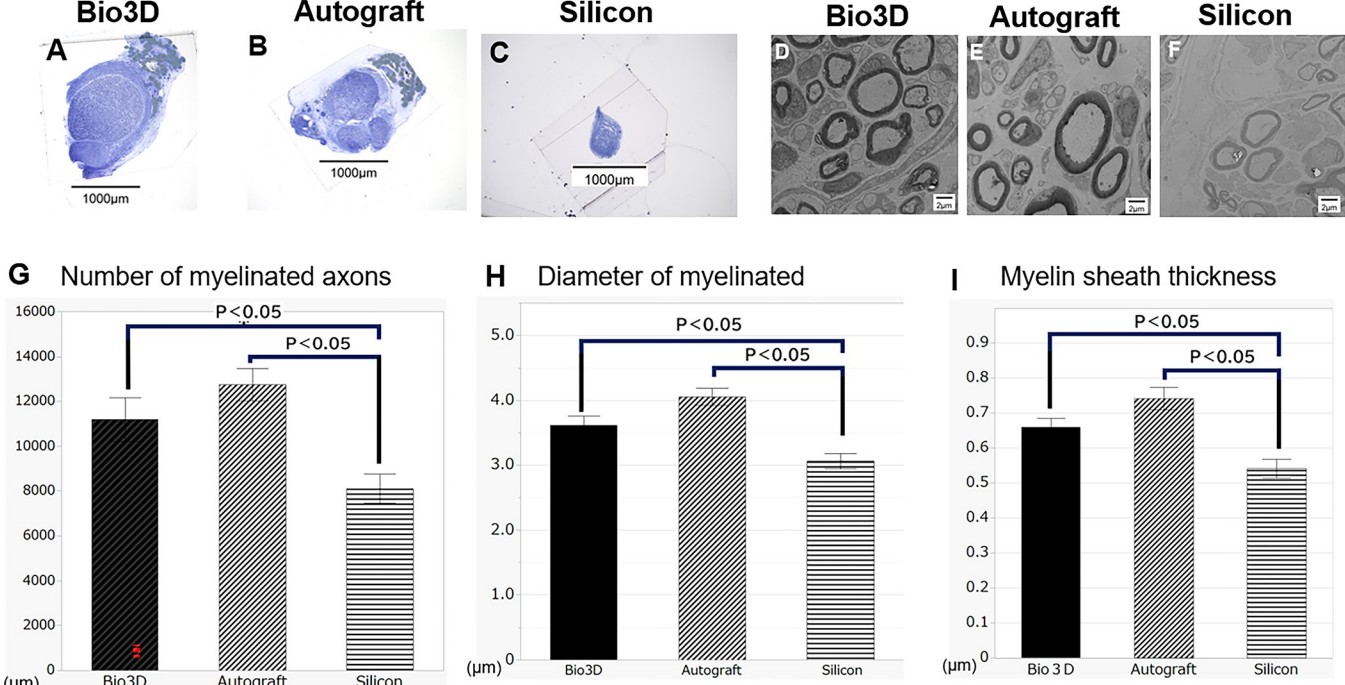

**Fig 4.** (A-C) Semi-thin transverse sections stained with toluidine blue (scale bar = 1000 μm): (A) Bio 3D group, (B) autograft group, (C) silicone group. (D-F) Ultrathin transverse sections observed with transmission electron microscopy (scale bar = 2 μm): (D) Bio 3D group, (E) autograft group, (F) silicone group. The regenerated nerves of the Bio 3D group were larger than those of the silicone group and similar to those of the autograft nerve group. (G) Number of myelinated axons, (H) diameter of myelinated axons, and (I) myelin sheath thickness. Significant differences were observed between the Bio 3D group and the silicone group, and between the autograft group and the silicone group in all three parameters. Error bars represent the standard error.

The mean number of myelinated axons was 11,201 ± 980 in the Bio 3D group, 12,750 ± 720 in the autograft group, and 8117 ± 646 in the silicone group. Significant differences were observed between the Bio 3D group and the silicone group, and between the autograft group and the silicone group (p = 0.0201 and p = 0.0003, respectively) (Fig 4G).

The mean diameter of myelinated axons was 3.61 ± 0.15 μm in the Bio 3D group, 4.06 ± 0.13 μm in the autograft group, and 3.07 ± 0.12 μm in the silicone group (Fig 4H). The average diameter of myelinated axons was significantly greater in the Bio 3D and autograft groups than in the silicone group (p = 0.0124 and p < 0.001, respectively); there was no significant difference between the Bio 3D and autograft nerve groups.

The mean myelin sheath thickness was 0.66 ± 0.03 μm in the Bio 3D group, 0.74 ± 0.03 μm in the autograft group, and 0.54 ± 0.03 μm in the silicone group (Fig 4I). The myelin sheath thickness of the Bio 3D and autograft groups was significantly greater than that of the silicone group (p = 0.0115 and p < 0.001, respectively) and there was no significant difference between the Bio 3D group and the autograft group.

## Mononuclear cell infiltration

In the Bio 3D, autograft, and allograft groups, HE staining of the regenerated nerves was performed 1 week after surgery and the number of mononuclear cells infiltrating on these sides was counted (Fig 5A–5C). The average number of monocytes per field was 238 ± 19 in the Bio

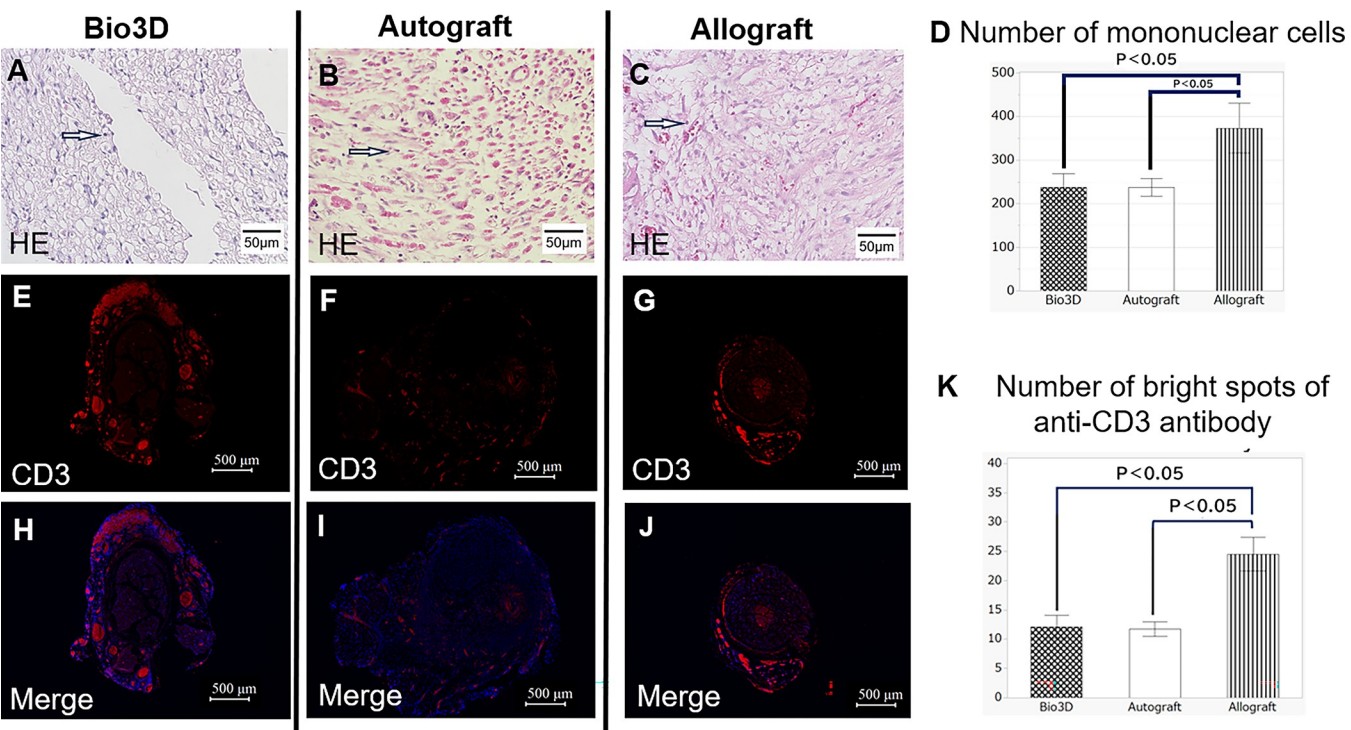

**Fig 5. Regenerated nerve 1 week after surgery, illustrating nerve rejection.** (A-C) HE staining of the transverse section in each of the three groups (scale bar = 50 μm). The infiltrative mononuclear cells are indicated by white arrows. We counted infiltrative mononuclear cells in the nerves and evaluated the inflammatory response. (D) Number of mononuclear cells infiltrating the nerves within the view field on HE staining. There were significantly lower counts in the Bio 3D group and the autograft group than in the allograft group. Error bars represent the standard error. (E-G) Anti-CD3 antibody staining of the transverse section in each of the three groups (scale bar = 500 μm). We considered the area indicated by the arrow below to be infiltrative mononuclear. (H-J) Anti-CD3 antibody and DAPI staining of the transverse section in each of the three groups (scale bar = 500 μm). Perineural staining in the Bio 3D group and the allograft group was observed as staining around the edge of the nerve. (K) Number of bright spots of anti-CD3 antibody. There were significantly lower counts in the Bio 3D group and the autograft group than in the allograft group. Error bars represent the standard error.

3D group, 238 ± 18 in the autograft group, and 374 ± 30 in the allograft group, with these counts being significantly smaller in the Bio 3D and autograft groups compared with the allograft group (p = 0.0004 and p = 0.0004, respectively) (Fig 5D).

## Immunohistochemistry

In the Bio 3D, autograft, and allograft groups, immunofluorescent staining of the regenerated nerves with anti-CD3 antibody was performed 1 week after surgery. Immunofluorescence staining with anti-CD3 antibody revealed a red stained area around the nerves in the Bio 3D group and the allograft group (Fig 5E–5J). The mean number of bright spots per field under high magnification was 12.1 ± 2.0 in the Bio 3D group, 11.7 ± 1.2 in the autograft group, and 24.5 ± 2.9 in the allograft group (Fig 6J), with these counts being significantly smaller in the

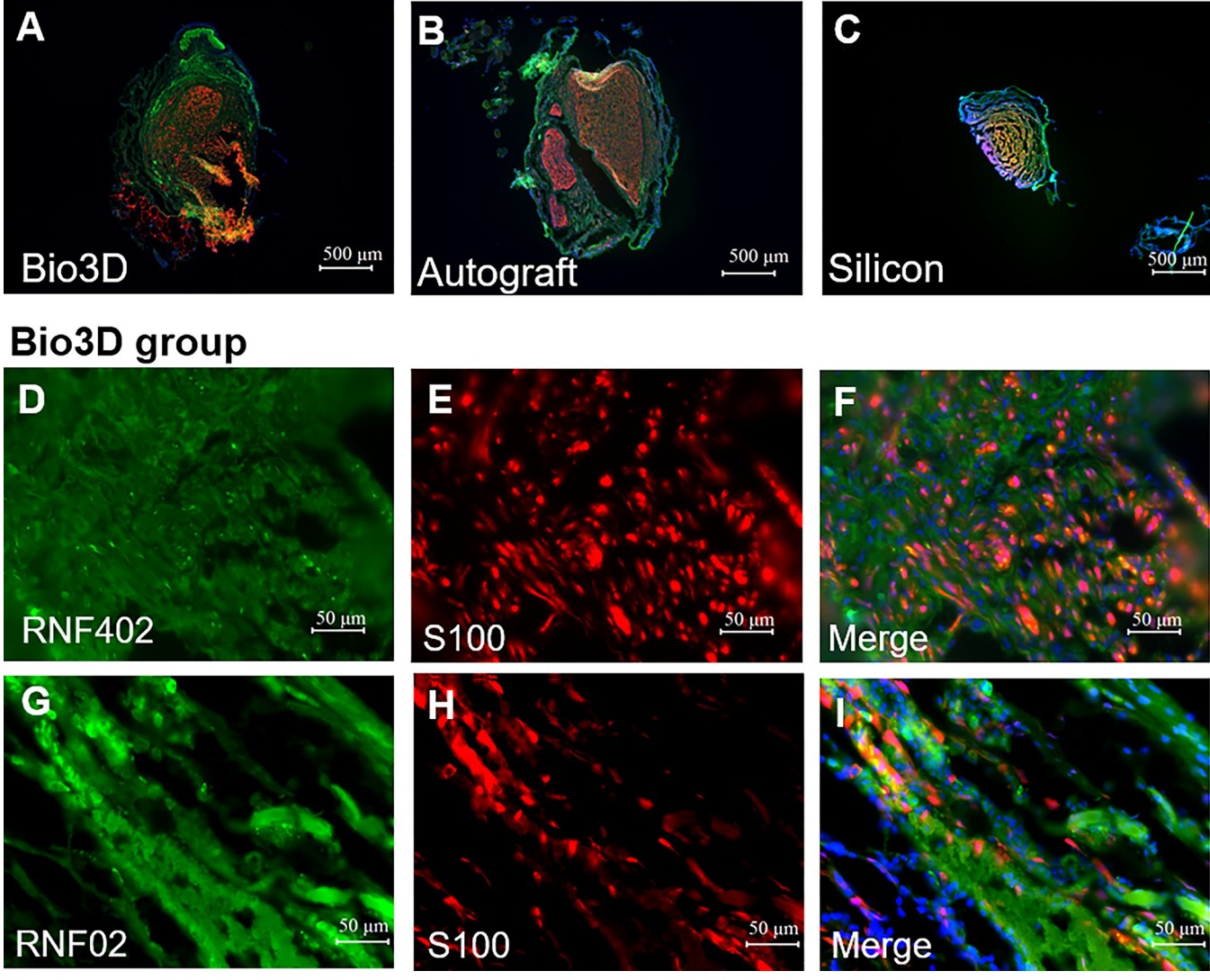

**Fig 6. Immunohistochemical staining of the regenerated nerve 8 weeks after surgery showing nerve recovery.** RNF402 is stained green, S100 is stained red, and the nucleus is stained blue using DAPI (A-C). The upper row shows merged images (RNF402/S100/DAPI) of the transverse section in each of the three groups (scale bar = 500 μm). (D-F) The middle row shows images of the transverse section in the Bio 3D group (scale bar = 50 μm). (G-I) The lower row shows images of the longitudinal section in the Bio 3D group (scale bar = 50 μm). Color development of RNF402 and S100 suggests the presence of nerve tissue.

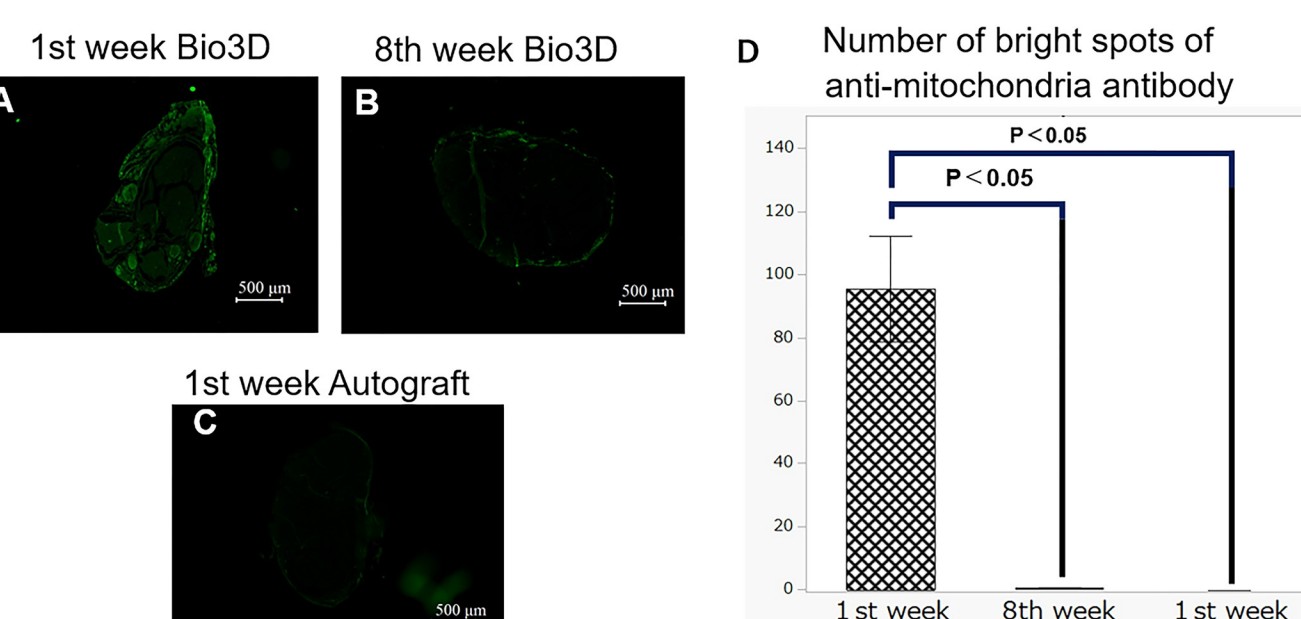

**Fig 7. Immunohistochemical staining to evaluate the change in the transplanted bio 3D conduit within the rat were between 1 week and 8 weeks after surgery.** (A-C) Anti-mitochondria antibody staining of the transverse section in each of the three groups (scale bar = 500 μm): (A) week 1, Bio 3D group, (B) week 8, Bio 3D group, (C) week 1, autograft group. (D) There were significantly greater values in the week 1 Bio 3D group than in the week 8 Bio 3D group and the week 1 autologous group. Error bars represent the standard error.

Bio 3D and autograft groups than in the allograft group (p = 0.0006 and p = 0.0009, respectively) (Fig 5K).

Expression of S-100 and RNF402, which indicates the presence of Schwann cells and nerve fibers, respectively, in the regeneration region, was observed in the central region of regenerated nerves in the Bio 3D, autograft, and silicone groups (Fig 6A–6C). Both transverse and longitudinal sections with S-100 and RNF402 staining are shown (Fig 6D–6F and 6G–6I, respectively).

In anti-mitochondria antibody immunofluorescence staining, perineural staining in the Bio 3D group 1 week after surgery was observed around the edge of the nerve; at 8 weeks after surgery, less staining was observed in the same region (Fig 7A–7C). The mean number of bright spots per field under high power was 95.6 ± 16.8 in the Bio 3D group 1 week after surgery, 0.6 ± 0.1 in the Bio 3D group 8 weeks after surgery, and 0 in the autograft nerve group 1 week after surgery (Fig 7D), with these values being significantly greater 1 week after surgery compared with 8 weeks after surgery in the Bio 3D and autograft groups 1 week after surgery (p < 0.01, respectively).

## Plasma cytokines

The concentration of IL-2 was 84.5 ± 41.8, 142.6 ± 37.1, and 122.6 ± 83.2 pg/mL and that of IL-10 was 87.6 ± 14.3, 113.5 ± 18.2, and 124.8 ± 15.8 pg/mL in the Bio 3D, autograft, and allograft groups, respectively. The concentration of IFN-γ was 147.4 ± 66.2, 178.1 ± 118.5, and 2109.7 ± 801.0 pg/mL in the Bio 3D, autograft, and allograft groups, respectively, and the concentration of TNF-α was 15.4 ± 2.7, 17.6 ± 2.0, and 15.9 ± 5.3 pg/mL in the three groups,

respectively. There were no significant differences in cytokine concentrations among the three groups (Supporting information).

## Discussion

In this study, we evaluated nerve regeneration using a Bio3D conduit derived from UC-MSCs; an autologous transplant was used as a positive control and a silicone tube was used as a negative control. Autologous nerves are excellent as crosslinking materials because they contain a scaffold structure, extracellular matrix, and vascular network for nerve regeneration. Many studies on artificial nerves aim for the artificial nerve to be comparable to the autologous nerve, so we also used the autologous nerve group as a positive control [14]. Scaffolds play an important role in nerve regeneration. The purpose of this study was to compare cross-linked materials for nerve regeneration evaluation. Thus, a silicone conduit with a luminal structure that prevents scar invasion but does not contain growth factors was used as a negative control [15].

In this study, we showed that nerve regeneration in the Bio 3D group outperformed that of the silicone group, with statistically significant differences in AoA, number of myelinated axons, myelinated axon diameter, and myelin thickness. We found no significant difference in the pinprick test, toe-spread test, or DT. However, due to the nature of the pinprick test and DT, scores may change depending on the physical condition and alertness of the rat. Additionally, the Bio 3D group tended to have better results than the silicone group in our study population of 15 animals, and it is possible that we would have found statistical differences between the groups with a larger population size. There were no significant differences in CMAP, NCV, or wet weight of the tibialis anterior muscle between the Bio 3D group and the silicone group. We believe this lack of significant findings in the electrophysiological tests is because the waveform is small at 8 weeks after surgery. With long-term follow-up observation, nerve regeneration progress and the accuracy of electrophysiological testing itself would be expected to improve.

Comparing the Bio 3D and autograft groups, the AoA results indicated significantly better nerve recovery in the Bio 3D group than in the autograft group, and there was no significant difference in other parameters. Fluorescent immunostaining revealed the presence of the neurofilament marker RNF402 and the Schwann cell marker S-100 in the Bio 3D group, suggesting that the transplanted UC-MSCs were replaced by neural tissue during regeneration.

Bojanic et al. previously reported that human UC-MSC transplantation is an effective approach for peripheral nerve regeneration [16]. Umbilical cords can be conveniently collected from postnatal tissue using a noninvasive method and have a high proliferation ability, making them advantageous in terms of material procurement [17]. UC-MSCs are known to have a greater paracrine effect than bone marrow–derived MSCs or adipose-derived MSCs, and are presumed to be superior for nerve regeneration [15]. Although MSCs have long been known for their ability to repair damaged tissue through direct differentiation, they also have an additional function: secreting trophic factors that enable the recruitment of endogenous stem/progenitor cells. UC-MSCs can also secrete brain-derived neurotrophic factor (BDNF) and increase the expression levels of local neurotransmitters such as BDNF and neurotrophin-3 (NTF3). It is also thought that UC-MSCs promote recovery of nerve regeneration [18–20].

In this study, we used a xenograft model in which human-derived tissue was used as a Bio 3D conduit to be transplanted into rats without immunosuppression. MSCs have been shown to lack expression of major histocompatibility complex II [21]. Their immunosuppressive properties are thought to enable xenotransplantation. In the second experiment, to evaluate the acute rejection reaction of a bio 3D conduit transplantation, we used an autologous group that caused relatively no response and an allograft group, which would be assumed to cause

rejection reactions, as controls. Because UC-MSCs have immune tolerance, we hypothesized that the effect of the Bio 3D group would be intermediate between that of the autologous group and the allograft group (in which nerve segments harvested from Brown Norway rats were transplanted to Lewis rats). To demonstrate this, we performed histological evaluation of the sciatic nerve and measured cytokine concentrations in blood 1 week after surgery. Monocyte infiltration peaks within 1 day after surgery and T lymphocyte infiltration begins 3 days after surgery and peaks between 14 and 28 days [22]. Due to the above-mentioned time-of-onset and sampling timing issues, we decided to evaluate the nerve rejection 1 week after surgery. The results of monocyte infiltration by HE staining and immunofluorescent staining of anti-CD3 antibody (a T cell marker) showed that immunoreactivity was lower in the Bio 3D group than in the allogeneic transplant group. No significant differences were found in plasma cytokine concentrations between the allograft group and the comparison group, suggesting that the experimental model itself had little effect on the whole body. In the future, human allogeneic UC-MSCs will be transplanted into nerve-damaged patients, and we can expect even better nerve regeneration than in this experimental xenograft model [9,23].

Previous studies reported that the number of MSCs decreases in the body after transplantation. MSCs can undergo apoptosis when administered to the body, and it has been reported that the majority of intravenously injected MSCs do not survive 1 week [24–26]. In our study, fluorescent immunostaining with an anti-human mitochondrial antibody revealed some stained UC-MSCs around the regenerated nerves 8 weeks after surgery. Considering past research results, it is assumed that UC-MSCs will gradually disappear. In this study, UC-MSCs were delivered directly to the nerve defect area in the form of a Bio3D conduit to provide a site for nerve regeneration and maintain a luminal structure that prevents scar invasion. Furthermore, the gradual disappearance of UC-MSCs allowed a period during which MSCs could secrete growth factors, neurotrophic factors, and cytokines. We assume that a system that delivers these factors to the living organism via a Bio3D conduit would be effective.

This study has several limitations. First, due to issues with UC-MSC procurement, this study was conducted with a relatively small number of animals. Second, the evaluation period for nerve regeneration was only 8 weeks, so long-term results are not yet available. Third, it has been reported that when the interneural gap is 10 mm and the conduit is silicone, nerve regeneration is considerably reduced [27]. Therefore, demonstrating regeneration in rats with an interneural gap of at least 10 mm is needed to demonstrate the usefulness of this approach in actual clinical practice. To achieve this, we will need to demonstrate that the Bio 3D conduits derived from UC-MSCs can be stacked at a distance of 10 mm or more. Fourth, the immunochemical studies in our research were not complete. We performed staining with anti-CD3 to evaluate the rejection response to a Bio 3D conduit transplantation, but staining with cytokines such as IFN-γ and IL-17 was not performed [28]. In addition, anti-human mitochondrial antibodies were used to evaluate the survival of human tissue 8 weeks after transplantation but HLA immunohistochemistry was not tested. Finally, we did not compare the UC-MSCs with the Bio3D conduit materials (fibroblasts, bone marrow stromal cells, induced pluripotent stem cells) that we have researched previously [5,7,13].

## Conclusions

The UC-MSC–derived Bio 3D conduit was superior to the silicone conduit and achieved nerve regeneration close to that of the autologous graft, suggesting the effectiveness of UC-MSCs in Bio 3D conduits. We observed greater suppression of the rejection reaction in the Bio 3D group than in the allograft group. Although we used a xenograft model in this study, the rejection reaction was low due to the characteristics of UC-MSCs. In terms of

UC-MSC residual evaluation, UC-MSCs in Bio 3D conduits remained but decreased in number between 1 and 8 weeks after transplantation.

## Supporting information

**S1 File. Blood concentrations (pg/mL) of (A) interleukin (IL)-2, (B) IL-10, (C) interferon (IFN)-γ, and (D) tumor necrosis factor (TNF)-α in the first week after surgery.** There was no significant difference in cytokine concentrations among the three groups.
(PPTX)

## Acknowledgments

We thank Keiko Furuta, Tatsuya Katsuno, and Haruyasu Kohda (Division of Electron Microscopic Study, Center for Anatomical Studies, Graduate School of Medicine, Kyoto University) for technical assistance in electron microscopy.

## Author Contributions

**Conceptualization:** Terunobu Iwai, Ryosuke Ikeguchi, Tomoki Aoyama, Shizuka Akieda, Tokiko Nagamura-Inoue, Fumitaka Nagamura, Koichi Nakayama, Shuichi Matsuda.

**Data curation:** Takashi Noguchi, Kazuaki Fujita, Yudai Miyazaki, Tokiko Nagamura-Inoue.

**Formal analysis:** Koichi Yoshimoto, Daichi Sakamoto, Kazuaki Fujita, Tokiko Nagamura-Inoue.

**Funding acquisition:** Ryosuke Ikeguchi, Tomoki Aoyama.

**Investigation:** Terunobu Iwai.

**Methodology:** Koichi Yoshimoto, Daichi Sakamoto, Kazuaki Fujita, Yudai Miyazaki, Tokiko Nagamura-Inoue.

**Supervision:** Ryosuke Ikeguchi, Shizuka Akieda, Fumitaka Nagamura, Koichi Nakayama, Shuichi Matsuda.

**Writing – original draft:** Terunobu Iwai.

**Writing – review & editing:** Ryosuke Ikeguchi, Tomoki Aoyama, Tokiko Nagamura-Inoue, Shuichi Matsuda.

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
