## [Decision Letter · Decision Letter 0]

7 May 2024

PONE-D-24-05750Nerve regeneration using a Bio 3D conduit derived from umbilical cord–derived mesenchymal stem cells in a rat sciatic nerve defect modelPLOS ONE

Dear Dr. Ikeguchi,

Thank you for submitting your manuscript to PLOS ONE. After careful consideration, we feel that it has merit but does not fully meet PLOS ONE’s publication criteria as it currently stands. Therefore, we invite you to submit a revised version of the manuscript that addresses the points raised during the review process.

We look forward to receiving your revised manuscript.

Kind regards,

Atsushi Asakura, Ph.D

Academic Editor

PLOS ONE

Journal Requirements:

3. We notice that your supplementary figures are included in the manuscript file. Please remove them and upload them with the file type 'Supporting Information'. Please ensure that each Supporting Information file has a legend listed in the manuscript after the references list.

Reviewers' comments:

Reviewer's Responses to Questions

**Comments to the Author**

1. Is the manuscript technically sound, and do the data support the conclusions?

Reviewer #1: No

Reviewer #2: Partly

2. Has the statistical analysis been performed appropriately and rigorously? 

Reviewer #1: No

Reviewer #2: Yes

3. Have the authors made all data underlying the findings in their manuscript fully available?

Reviewer #1: No

Reviewer #2: Yes

4. Is the manuscript presented in an intelligible fashion and written in standard English?

Reviewer #1: No

Reviewer #2: Yes

5. Review Comments to the Author

Reviewer #1: In this study, the authors utilized a Kenzan-based approach to engineer 3D biological conduits made from umbilical cord-derived mesenchymal stem cells (UC-MSCs). The therapeutic efficacy of these engineered bio 3D conduits was evaluated in vivo using a rat model of sciatic nerve transection. This research presents an interesting application of Kenzan technology, but its impact could be further amplified by addressing several areas of concern:

1. The Abstract does not explain the fabrication process of the conduit.

2. The Introduction provides limited background information. MSCs have been previously used in artificial nerve conduits to enhance peripheral nerve regeneration. The study should clarify the key differences between its approach and those documented in existing literature, particularly addressing the limitations of MSC-incorporated conduits that this study overcomes.

3. The description of the current methods is overly simplistic, lacking the detail necessary to accurately replicate the results of this study. For example, the methodology for inducing MSC spheroid formation and the specific parameters utilized in the Kenzan technique should be thoroughly detailed.

4. The bio 3D conduits require comprehensive characterization.

5. Including a healthy control group in the pinprick and toe-spread tests, kinematic analysis, and electrophysiological studies would significantly enhance the manuscript's quality.

6. For the analyses of compound muscle action potential and nerve conduction velocity, presenting representative waveforms triggered by stimulation could improve the quality of the manuscript.

7. In Figure 3D, the weight of the tibialis anterior muscle in the affected limb should be normalized to that of the healthy contralateral limb for better objectivity.

8. Figures 5A-C should include labels for the colors and their corresponding targets.

9. A typographical error "Merage" was identified in Figure 5F.

10. The sequence of presenting data in the Results section is unconventional, discussing Figures 6A-6F, then Figure 7, followed by Figures 6G-6K.

11. The results from the plasma cytokine analysis should be incorporated into one of the figures in the manuscript.

12. The Discussion section is currently too brief and lacks depth. A more comprehensive analysis and interpretation of the findings are warranted.

13. The rationale for using two different strains of rats in a study that employs human-derived UC-MSCs is unclear.

14. The manuscript should clearly define the symbols used to denote statistical analysis results in the figure captions.

15. The manuscript would benefit from improvements in English language use and overall flow.

Reviewer #2: This article is to research a bio 3D conduit derived from umbilical cord–derived mesenchymal stem cells for peripheral nerve injury. The authors seem to aim a clinical application of this conduit. It is very important and timely to develop the strategy to repair peripheral nerve injury by regenerative medicine, thus the authors’ concept is reasonable and scientific in this meaning. However, some major deficiencies are seen in this article. These are described below:

Major points:

1. The authors should put a negative control group. For all meanings, this study completely lacks the negative control group, such as rats which were only undergone sciatic nerve cut (5mm gap). The reviewer could not assess all data whether a 3D conduit effect and an autografting effect, the differences of allo-reactive reaction or foreign body reaction and normal peripheral regeneration or not.

2. The authors should explain the reason why actual function (ex. Pinprick test/ toe-spread test, DT, even though AoA showed significance) did not show differences even though the regenerative histology showed differences. According to these data, the conclusion resulted that 3D conduit accelerate peripheral nerve injury histologically, however, did not affect functional recovery. Nobody will agree that 3D conduit should be used for clinical application.

3. Did Figure 1D and 4A show 3D conduit or sciatic nerve itself? Concerning Figure 1C, sciatic nerve seemed inside silicon tube (the small knots could be pointed out lower part of Figure 1D. Does it mean that the whitish sheath remained?) Was a 3D conduit absorbed after 8 weeks? Did the authors investigate HLA immunohistochemistry to investigate which tissue was regenerated?

4. The conclusion of xeno- or allo-transplantation immunity was premature. At first, where were infiltrative mononuclear cells in Figure 6G, H and I? The reviewer completely confused whether the infiltrative mononuclear cells existed or not in these figures. Secondly, why the authors only showed 7 post-operative day? Thirdly, why the authors investigated pan-T (CD3) only? The data must differ from infiltrated monocytes and pan-T cells.

5. Authors must show the citations if the authors stated that “UC-MSCs are known to have a greater paracrine effect than bone marrow–derived MSCs or adipose derived MSCs, and are presumed to be superior for nerve regeneration” in Discussion section.

Minor point:

1. Figure 6K is poor.

6. PLOS authors have the option to publish the peer review history of their article (what does this mean?). If published, this will include your full peer review and any attached files.

Reviewer #1: No

Reviewer #2: No

---

## [Author Response · Author response to Decision Letter 0]

23 Jun 2024

Reviewer #1

Thank you very much for reviewing our manuscript. We agree with your comments and suggestions and have revised our manuscript accordingly.

1. The Abstract does not explain the fabrication process of the conduit.

Thank you for your valuable comment. According to your suggestion, we added the sentence to describe the fabrication process of the conduit in the Abstract (line 37-38).

2. The Introduction provides limited background information. MSCs have been previously used in artificial nerve conduits to enhance peripheral nerve regeneration. The study should clarify the key differences between its approach and those documented in existing literature, particularly addressing the limitations of MSC-incorporated conduits that this study overcomes.

Thank you for your valuable comment. According to your suggestion, we added the sentence to describe the key differences from our previous studies (line72-82).

3. The description of the current methods is overly simplistic, lacking the detail necessary to accurately replicate the results of this study. For example, the methodology for inducing MSC spheroid formation and the specific parameters utilized in the Kenzan technique should be thoroughly detailed.

Thank you for your instruction. According to your instruction, we added the sentences to explain the methods (line103 -116). 

4. The bio 3D conduits require comprehensive characterization.

Thank you for your valuable comment. According to your suggestion, we added the sentence to describe the characterization of the Bio 3D conduits (line115-116).

5. Including a healthy control group in the pinprick and toe-spread tests, kinematic analysis, and electrophysiological studies would significantly enhance the manuscript's quality.

Thank you for your instruction. In the pinprick test and toe-spread tests, the healthy group received a Grade 3. Regarding DT, the healthy control group has a value close to 0. Regarding AoA, the surgical treatment group takes a negative value, whereas the healthy control group takes a positive value. In electrophysiological studies, evaluation was performed based on the ratio of the affected side to the healthy side.

According to your suggestion, we added the sentence to describe the control group in the Discussion and reference 14 (line39-403) . 

6. For the analyses of compound muscle action potential and nerve conduction velocity, presenting representative waveforms triggered by stimulation could improve the quality of the manuscript.

Thank you for your instruction. According to your instruction, we added the waveforms in Figure 2 

7. In Figure 3D, the weight of the tibialis anterior muscle in the affected limb should be normalized to that of the healthy contralateral limb for better objectivity.

Thank you for your valuable comment. According to your suggestion, we added the ratio of the wet muscle weight of the operated side to that of the heathy side. We also added the sentences to describe the methods and results (line 209-210) (line 326-329). We revised Figure 3.

8. Figures 5A-C should include labels for the colors and their corresponding targets.

Thank you for your instruction. According to your instruction, we added the labels for the colors and their corresponding targets in Figure 5. We also revise the figure legend of Figure 5.

9. A typographical error "Merage" was identified in Figure 5F.

We appreciate your pointing out the mistake. We replace "Merage" with “Merge” in Figure 5.

10. The sequence of presenting data in the Results section is unconventional, discussing Figures 6A-6F, then Figure 7, followed by Figures 6G-6K.

Thank you for your instruction. According to your instruction, we put the order of Mononuclear cell infiltration ~ Immunohistochemistry to make it easier to understand and arrange the photos and graphs of Figure 6.

11. The results from the plasma cytokine analysis should be incorporated into one of the figures in the manuscript.

Thank you for your instruction. According to your instruction, we have described cytokines in Figure 8. However, the editor recommend us upload them as Supporting Information.

12. The Discussion section is currently too brief and lacks depth. A more comprehensive analysis and interpretation of the findings are warranted.

Thank you for your instruction. According to your instruction, we added the sentences in the Discussion section (line 394-412) (line 439-451).

13. The rationale for using two different strains of rats in a study that employs human-derived UC-MSCs is unclear　to evaluate acute graft rejection, in the second experiment, the Bio 3D group was compared with the autograft group and the allograft group (in which nerve segments harvested from Brown Norway rats were transplanted to Lewis rats) 1 week after surgery. 

Thank you for your instruction. According to your instruction, we added the sentences in the Discussion section to explain the reason of using two different strains of rats (line 439-445).

14. The manuscript should clearly define the symbols used to denote statistical analysis results in the figure captions.

Thank you for your instruction. According to your instruction, we added replace the symbols with the statistical analysis results in figures.

15. The manuscript would benefit from improvements in English language use and overall flow.

Thank you for your instruction. According to your instruction, we asked again the company of proofreading to improve English.

Reviewer #2:

Thank you very much for reviewing our manuscript. We agree with your comments and suggestions and have revised our manuscript accordingly.

1. The authors should put a negative control group. For all meanings, this study completely lacks the negative control group, such as rats which were only undergone sciatic nerve cut (5mm gap). The reviewer could not assess all data whether a 3D conduit effect and an autografting effect, the differences of allo-reactive reaction or foreign body reaction and normal peripheral regeneration or not.

Thank you for your valuable comment. Scaffolds play an important role in nerve regeneration. The purpose of this study was to compare cross-linked materials for nerve regeneration evaluation, and a silicone group that has a luminal structure that prevents scar invasion but does not contain growth factors was used as a negative control.

According to your suggestion, we added the sentences in conclusion as our study limitation and reference 15 (line 394-403).

2. The authors should explain the reason why actual function (ex. Pinprick test/ toe-spread test, DT, even though AoA showed significance) did not show differences even though the regenerative histology showed differences. According to these data, the conclusion resulted that 3D conduit accelerate peripheral nerve injury histologically, however, did not affect functional recovery. Nobody will agree that 3D conduit should be used for clinical application.

Thank you for your instruction. According to your instruction, we added the explanations in Discussion (line 406-412).

3. Did Figure 1D and 4A show 3D conduit or sciatic nerve itself? Concerning Figure 1C, sciatic nerve seemed inside silicon tube (the small knots could be pointed out lower part of Figure 1D. Does it mean that the whitish sheath remained?) Was a 3D conduit absorbed after 8 weeks? Did the authors investigate HLA immunohistochemistry to investigate which tissue was regenerated?

Thank you for your valuable comment. In Figure 1D, most of the nerve appears to be regenerated sciatic nerve. In Figure 4A, we replaced the figure to represent that most of the nerves appear to be regenerated in sciatic nerve and 3D conduit absorbed after 8 weeks. 

According to your comments, we add the sentences in Discussion (line 479-484).

4. The conclusion of xeno- or allo-transplantation immunity was premature. At first, where were infiltrative mononuclear cells in Figure 6G, H and I? The reviewer completely confused whether the infiltrative mononuclear cells existed or not in these figures. Secondly, why the authors only showed 7 post-operative day? Thirdly, why the authors investigated pan-T (CD3) only? The data must differ from infiltrated monocytes and pan-T cells.

Thank you for your valuable comment. According to your comments, we added the arrow to show infiltrative mononuclear cells (Figure5). We also add the explanation in figure legend of Figure 5.　We also add the explanation in the Discussion section and reference 22 and 28 (line445 -451) (line 479-487).

5. Authors must show the citations if the authors stated that “UC-MSCs are known to have a greater paracrine effect than bone marrow–derived MSCs or adipose derived MSCs, and are presumed to be superior for nerve regeneration” in Discussion section.

Thank you for your valuable comment. According to your suggestion, we added reference 15 (line 430).

Minor point:

1. Figure 6K is poor.

Thank you for your instruction. According to your instruction, we modified the figure (Figure 5D in revised version).

---

## [Editor Report · Decision Letter 1]

6 Sep 2024

Nerve regeneration using a Bio 3D conduit derived from umbilical cord–derived mesenchymal stem cells in a rat sciatic nerve defect model

PONE-D-24-05750R1

Dear Dr. Ikeguchi,

We’re pleased to inform you that your manuscript has been judged scientifically suitable for publication and will be formally accepted for publication once it meets all outstanding technical requirements.

Kind regards,

Atsushi Asakura, Ph.D

Academic Editor

PLOS ONE
---

## [Editor Report · Acceptance letter]

14 Oct 2024

PONE-D-24-05750R1 

PLOS ONE

Dear Dr. Ikeguchi, 

I'm pleased to inform you that your manuscript has been deemed suitable for publication in PLOS ONE. Congratulations! Your manuscript is now being handed over to our production team.

Kind regards, 

on behalf of

Dr. Atsushi Asakura 

Academic Editor

PLOS ONE